# Effects of steroid therapy in patients with severe fever with Thrombocytopenia syndrome: A multicenter clinical cohort study

Sook In Jung[1☯], Ye Eun Kim[2☯], Na Ra Yun[3], Choon-Mee Kim[4], Dong-Min Kim[3]*, Mi Ah Han[4], Uh Jin Kim[1], Seong Eun Kim[1], Jieun Kim[5], Seong Yeol Ryu[6], Hyun ah Kim[6], Jian Hur[7], Young Keun Kim[8], Hye Won Jeong[9], Jung Yeon Heo[10], Dong Sik Jung[11], Hyungdon Lee[12], Kyungmin Huh[13], Yee Gyung Kwak[14], Sujin Lee[15], Seungjin Lim[15], Sun Hee Lee[16], Sun Hee Park[17], Joon-Sup Yeom[18], Shin-Woo Kim[19], In-Gyu Bae[20], Juhyung Lee[21], Eu Suk Kim[22], Jun-Won Seo[3]

1 Department of Internal Medicine, Chonnam National University Medical School, Gwangju, Republic of Korea, 2 Department of Nursing, College of Medicine, Chosun University, Gwangju, Republic of Korea, 3 Department of Internal Medicine, College of Medicine, Chosun University, Gwangju, Republic of Korea, 4 Department of Preventive Medicine, College of Medicine, Chosun University, Gwangju, Republic of Korea, 5 Department of Internal Medicine, College of Medicine, Hanyang University, Seoul, Republic of Korea, 6 Division of Infectious Diseases, Keimyung University Dongsan Medical Center, Daegu, Republic of Korea, 7 Department of Internal Medicine, Yeungnam University Medical Center, Daegu, Republic of Korea, 8 Department of Internal Medicine, Wonju College of Medicine, Yonsei University, Wonju, Republic of Korea, 9 Department of Internal Medicine, College of Medicine, Chungbuk National University, Cheongju, Republic of Korea, 10 Department of Infectious Diseases, School of Medicine, Ajou University, Suwon, Republic of Korea, 11 Department of Internal Medicine, College of Medicine, Dong-A University, Busan, Republic of Korea, 12 Department of Internal Medicine, College of Medicine, Hallym University, Chuncheon, Republic of Korea, 13 Division of Infectious Diseases, Department of Medicine, Samsung Medical Center, Seoul, Republic of Korea, 14 Department of Internal Medicine, Inje University Ilsan Paik Hospital, Goyang, Republic of Korea, 15 Department of Internal Medicine, College of Medicine, Pusan National University, Yangsan, Republic of Korea, 16 Department of Internal Medicine, Pusan National University School of Medicine, Pusan National University Hospital, Busan, Republic of Korea, 17 Division of Infectious Diseases, Department of Internal Medicine, College of Medicine, The Catholic University of Korea, Daejeon, Republic of Korea, 18 Department of Internal Medicine, College of Medicine, Yonsei University, Seoul, Republic of Korea, 19 Department of Internal Medicine, School of Medicine, Kyungpook National University, Daegu, Republic of Korea, 20 Department of Internal Medicine, College of Medicine, Gyeongsang National University, Jinju, Republic of Korea, 21 Department of Preventive Medicine, Jeonbuk National University Medical School, Jeonju, Republic of Korea, 22 Department of Internal Medicine, Seoul National University Bundang Hospital, Seongnam, Republic of Korea

☯ These authors contributed equally to this work.
* drongkim@chosun.ac.kr

**Data Availability Statement:** All relevant data are within the manuscript and its Supporting Information files.

## Abstract

### Background

Severe fever with thrombocytopenia syndrome (SFTS) is an acute, febrile, and potentially fatal tick-borne disease caused by the SFTS *Phlebovirus*. Here, we evaluated the effects of steroid therapy in Korean patients with SFTS.

### Methods

A retrospective study was performed in a multicenter SFTS clinical cohort from 13 Korean university hospitals between 2013 and 2017. We performed survival analysis using propensity score matching of 142 patients with SFTS diagnosed by genetic or antibody tests.

**Funding:** D-MK received grant number 2017-P23002-00 by the Korea Centers for Disease Control and Prevention (KCDC), http://www.kdca.go.kr. The funder had no role in study design, data collection and analysis, decision to publish, or preparation of the manuscript.

**Competing interests:** The authors have declared that no competing interests exist.

## Results

Overall fatality rate was 23.2%, with 39.7% among 58 patients who underwent steroid therapy. Complications were observed in 37/58 (63.8%) and 25/83 (30.1%) patients in the steroid and non-steroid groups, respectively ($P < .001$). Survival analysis after propensity score matching showed a significant difference in mean 30-day survival time between the non-steroid and steroid groups in patients with a mild condition [Acute Physiology and Chronic Health Evaluation II (APACHE II) score <14; 29.2 (95% CI 27.70–30.73] vs. 24.9 (95% CI 21.21–28.53], $P = .022$]. Survival times for the early steroid ($\leq 5$ days from the start of therapy after symptom onset), late steroid (>5 days), and non-steroid groups, were 18.4, 22.4, and 27.3 days, respectively ($P = .005$).

## Conclusions

After steroid therapy, an increase in complications was observed among patients with SFTS. Steroid therapy should be used with caution, considering the possible negative effects of steroid therapy within 5 days of symptom onset or in patients with mild disease (APACHE II score <14).

### Author summary

Severe fever with thrombocytopenia syndrome (SFTS) is an acute, febrile, and potentially fatal tick-borne disease caused by the SFTS *Phlebovirus*. Here, we evaluated the effects of steroid therapy in Korean patients with SFTS. We performed survival analysis using propensity score matching of 142 patients with SFTS diagnosed by genetic or antibody tests. In patients with SFTS, steroid therapy should be used with caution, considering the possible negative effects of steroid therapy within 5 days of symptom onset or in patients with mild disease (APACHE II score <14).

## Introduction

Severe fever with thrombocytopenia syndrome (SFTS) is an acute febrile disease caused by an SFTS virus (SFTSV) belonging to the *Phlebovirus* genus [1,2]. The infection occurs when a tick infected with SFTSV bites a person. *Haemaphysalis longicornis*, *Amblyomma testudinarium*, and *Ixodes nipponensis* are the vectors mediating this disease [3,4].

 SFTS was first identified in China in 2011 [5], and new patients have been registered annually in Korea since the confirmation of the first Korean patient in 2013 [6]. Since then, in Japan and Taiwan, SFTSV-infected patients or ruminants and ticks have been identified [7,8]. The 2011 and 2012 mortality rates for patients with SFTS in China were 6% (n = 123 deaths among 2,017 confirmed patients), whereas those in Korea and Japan exceeded 30% [9,10].

 Due to the lack of therapeutic agents, there is no standardized treatment for SFTS, and current treatment is based on symptomatic therapy. Short-term glucocorticoid therapy could be helpful in the treatment of encephalopathy in early-stage SFTS infection [11]. However, such beneficial treatment effects have only been reported in case studies or studies including a small number of patients.

 Few studies have assessed the clinical characteristics of patients with SFTS or analyzed the clinical characteristics related to the risk to death [12]. In addition, no systematic analyses have

investigated the effects of steroid therapy in patients with SFTS. This study evaluated the effects of steroid therapy by analyzing the epidemiologic and clinical characteristics of patients with SFTS in Korea and the therapeutic effects of steroid therapy using propensity score matching.

## Methods

### Ethics statement

Research was approved by the Ethics in Human Research Committee of Chosun University Hospital (IRB No. 2017-10-012) as the central coordinating center, and written informed consent was provided by all participants in the study.

### Study setting and participants

A multicenter SFTS clinical cohort study was established to identify the treatment status and analyze the effects of treatment in patients with SFTS at specific institutions. Retrospective studies were undertaken to analyze the epidemiological and clinical characteristics of patients with SFTS in 13 university hospitals in Korea after reviewing medical records. After receiving approval from the Institutional Review Board of each participating institution, patients with a confirmed diagnosis of SFTS were selected for data collection at each participating institution. The inclusion criteria were that 1) patients who admitted to hospitals from 2013 to 2017, and 2) those who confirmed SFTS by molecular or serology test for SFTSV.

A diagnosis of SFTS was made by the Korean Center for Disease Control (KCDC) [13], Korea Institute of Health and Environment, or Chosun University Hospital. Diagnosis was made on the basis of results from conventional, nested polymerase chain reaction (PCR) or real-time (RT)-PCR to detect viral RNA in the blood of patients with SFTS. In some cases, diagnosis was made via immunofluorescence assays for the SFTSV and confirmed if there was more than four-fold increase in SFTS IgM or IgG antibodies [14,15].

To exclude the effects of other diseases, all patients with co-infection for any other pathogen besides SFTSV were excluded. Epidemiologic data, clinical features, laboratory results, and treatment outcomes of patients with SFTS were collected after reviewing medical records and previously completed epidemiological forms.

### Statistical analysis

Results of categorical variables were expressed as frequencies and percentages, while those of continuous variables were presented as medians and interquartile ranges (IQRs). Analyses were performed using the Mann–Whitney U-test to compare continuous variables between the fatal and non-fatal patient groups and between the steroid and non-steroid therapy groups. For categorical variables, analyses were performed using Chi-square or Fisher's exact tests.

For comparative analysis of treatment outcomes, clinical laboratory results pre-steroid therapy and 48 hours post-steroid therapy were compared using the Wilcoxon signed-rank test. To identify 30-day mortality risk factors (from admission) in patients with SFTS, univariate and multivariate Cox proportional hazard regressions were conducted. The effects of various therapies on 30-day mortality risk in patients with SFTS were identified using a Cox regression model constructed by adjusting for two covariates; initial Acute Physiology and Chronic Health Evaluation (APACHE) II score [16], which represented disease severity, and hospitalization within 7 days of symptom onset, which may have affected the application of treatment.

To identify the effects of steroid therapy on 30-day survival time, a Kaplan-Meier survival analysis was conducted in the steroid and non-steroid groups after propensity score matching. Patients with severe SFTS were more likely to have received steroid therapy; therefore, owing

to the high probability of selection bias, corrections were made by propensity score matching using the nearest method without caliper. For the generation of propensity score-matched cohorts, confounding factors were identified using logistic regression for both groups and 1:1 matching was performed by the propensity score calculated using a logistic regression model to match variables in both groups. The final matching variables included age, sex, underlying comorbidities, initial APACHE II score, initial respiration rate, initial altered mental state, and initial intensive care unit (ICU) admission. All calculated *P* values were two-sided, and 95% confidence intervals were considered. Statistical analyses were performed using the Windows IBM SPSS software (version 24, IBM corp., NY, USA) and R software (version 3.6.1).

## Results

### Epidemiologic and clinical characteristics of patients with SFTS

A total of 142 patients had a confirmed diagnosis of SFTS between 2013 and 2017 by PCR test, and 12 of them were also diagnosed via more than 4 fold IgG antibody increase using IFA. No patient had a confirmed co-infection such as scrub typhus, leptospirosis, and hemorrhagic fever with renal syndrome by PCR tests. Each year, 3–75 patients were infected with SFTS, and the case fatality rate was 16.7%–66.7% (Fig 1). Of the 142 patients, 33 died (case fatality rate, 23.2%). The male-to-female ratios in the fatal and non-fatal groups were 45.5:54.5 and 52.3:47.7, respectively; however, this was not a statistically significant difference (*P* = .554) (Table 1).

The median age of the 142 patients was 68.5 years. A statistically significant difference in median age was observed between the non-fatal and fatal groups (67 vs. 75 years, *P* < .001). Of the 142 patients, 88 (62.0%) had confirmed comorbidities, including 64 (58.7%) and 24 (72.7%) patients in the non-fatal and fatal groups, respectively. Data on complications were examined in 141 of 142 patients. Of the 62 patients who experienced complications, ventilator

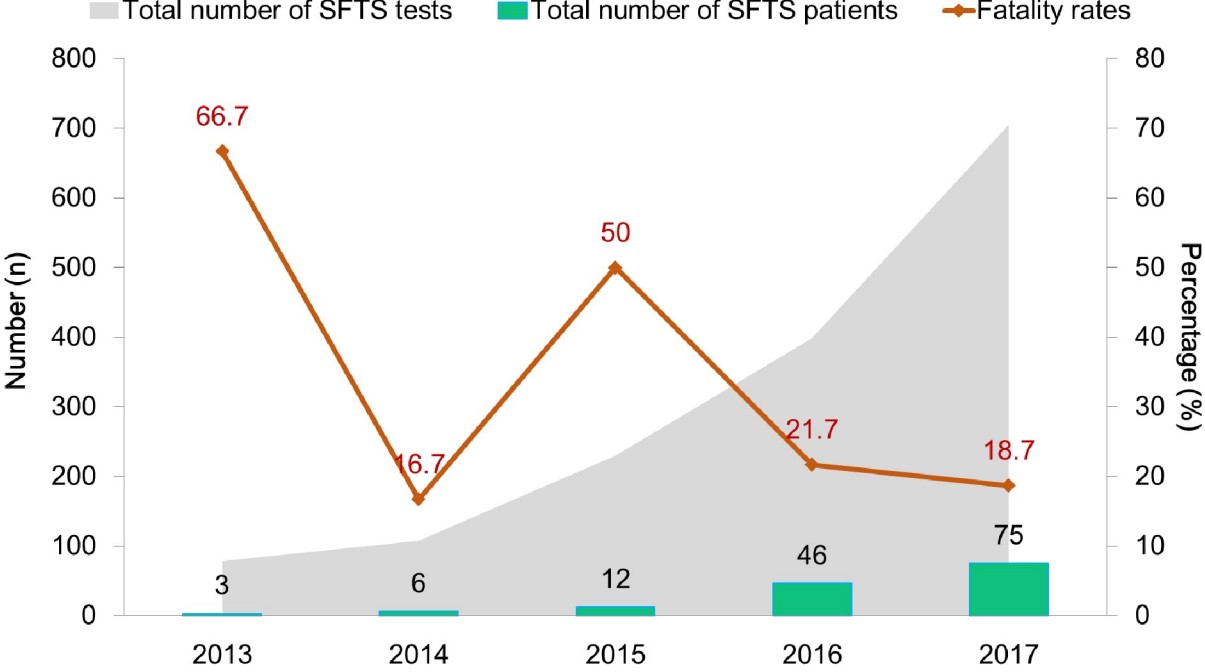

**Fig 1. Number of patients enrolled in multicenter cohorts with confirmed severe fever with thrombocytopenia syndrome (SFTS) and mortality between 2013 and 2017.**

**Table 1. General and Clinical Characteristics of Patients with SFTS in the Non-Fatal and Fatal Groups (2013–2017).**

| Characteristics | Non-Fatal | | Fatal[a] | | Total | | P Value[b] |
|---|---|---|---|---|---|---|---|
| | n = 109 | | n = 33 | | N = 142 | | |
| Age, years (median, IQR) | 67.0 | (59.0–73.0) | 75.0 | (67.5–81.5) | 68.5 | (61.0–75.3) | < .001 |
| Sex, Male | 57 | (52.3) | 15 | (45.5) | 72 | (50.7) | .554 |
| Comorbidity, total | 64 | (58.7) | 24 | (72.7) | 88 | (62.0) | .146 |
| Initial clinical manifestation[c] | | | | | | | |
| Fever | 97 | (89.0) | 24 | (77.4) | 121 | (86.4) | .033 |
| Chills | 70 | (64.2) | 11 | (35.5) | 81 | (57.9) | .005 |
| Myalgia | 53 | (48.6) | 7 | (22.6) | 60 | (42.9) | .008 |
| Cough | 10 | (9.2) | 4 | (12.9) | 14 | (10.0) | .492 |
| Nausea | 31 | (28.4) | 7 | (22.6) | 38 | (27.1) | .814 |
| Vomiting | 18 | (16.5) | 7 | (22.6) | 25 | (17.9) | .416 |
| Diarrhea | 31 | (28.4) | 11 | (35.5) | 42 | (30.0) | .366 |
| Altered mental state | 18 | (16.5) | 11 | (35.5) | 29 | (20.7) | .028 |
| Glasgow Coma Scale (median, IQR) | 15.0 | (14.0–15.0) | 15.0 | (11.0–15.0) | 15.0 | (14.0–15.0) | .203 |
| Vital sign at first clinic visit | | | | | | | |
| Body temperature (˚C) (median, IQR) | 38.2 | (37.1–38.6) | 38.0 | (37.1–38.7) | 38.0 | (37.1–38.6) | .842 |
| Respiration rate (/min) (median, IQR) | 20.0 | (20.0–20.0) | 20.0 | (20.0–22.0) | 20.0 | (20.0–20.0) | .013 |
| Symptom onset to admission (median days, IQR) | 5.0 | (3.0–7.0) | 4.0 | (3.0–6.0) | 5.0 | (3.0–7.0) | .254 |
| Initial APACHE II score (median, IQR) | 10.0 | (8.0–14.0) | 16.0 | (11.0–20.0) | 11.0 | (9.0–16.0) | < .001 |
| Prior antibiotic treatment | 39 | (36.8) | 16 | (48.5) | 55 | (39.6) | .230 |
| CRRT/hemodialysis | 5 | (4.6) | 11 | (33.3) | 16 | (11.3) | < .001 |
| ICU admission during hospitalization | 34 | (31.5) | 28 | (84.8) | 62 | (44.0) | < .001 |
| Steroids treatment | 35 | (32.1) | 23 | (69.7) | 58 | (40.8) | < .001 |
| Days from admission to first treatment (median, IQR) | 2.0 | (1.0–4.0) | 2.0 | (1.0–3.0) | 2.0 | (1.0–3.3) | .089 |
| Total days of steroids administration (median, IQR) | 5.0 | (3.0–9.0) | 4.0 | (2.0–6.0) | 4.5 | (3.0–7.5) | .077 |
| Steroid dose (median, IQR) [d] | | | | | | | |
| Dexamethasone (mg/day) | 15.0 | (10.0–16.0) | 15.0 | (10.0–20.0) | 15.0 | (10.0–18.0) | .658 |
| Prednisolone (mg/day) | 20.0 | (10.0-N/A) | 10.0 | N/A | 15.0 | (10.0–20.0) | .500 |
| Methylprednisolone (mg/day) | 125.0 | (57.5–1000.0) | 62.5 | (50.0–125.0) | 68.8 | (56.3–417.5) | .252 |
| Hydrocortisone (mg/day) | 200.0 | (100.0-N/A) | 200.0 | (200.0–300.0) | 200.0 | (200.0–300.0) | .183 |
| Complications, total[e] | 32 | (29.6) | 30 | (90.9) | 62 | (44.0) | < .001 |
| Meningoencephalitis | 6 | (5.6) | 3 | (9.1) | 9 | (6.4) | .437 |
| Mechanical ventilation | 12 | (11.1) | 21 | (63.6) | 33 | (23.4) | < .001 |
| Arrhythmia | 1 | (0.9) | 7 | (21.2) | 8 | (5.7) | < .001 |
| Pneumonia | 6 | (5.6) | 5 | (15.2) | 11 | (7.8) | .129 |
| Seizure | 3 | (2.8) | 1 | (3.0) | 4 | (2.8) | 1.000 |
| Rhabdomyolysis | 2 | (1.9) | 1 | (3.0) | 3 | (2.1) | .554 |
| MOD | 0 | (0.0) | 1 | (3.0) | 1 | (0.7) | N/A |
| Septic shock/sepsis | 2 | (1.9) | 9 | (27.3) | 11 | (7.8) | < .001 |
| Acute kidney injury | 6 | (5.6) | 6 | (18.2) | 12 | (8.5) | .034 |
| GI bleeding | 2 | (1.9) | 2 | (6.1) | 4 | (2.8) | .233 |
| DIC | 0 | (0.0) | 4 | (12.1) | 4 | (2.8) | N/A |

(*Continued*)

**Table 1.** (Continued)

| Characteristics | Non-Fatal n = 109 | | Fatal[a] n = 33 | | Total N = 142 | | P Value[b] |
|---|---|---|---|---|---|---|---|
| Others[f] | 7 | (6.5) | 6 | (18.2) | 13 | (9.2) | .078 |

Data are presented as no. (%) unless otherwise indicated. Additional information of general and clinical characteristics of patients with SFTS are provided in the S1 Table.

Abbreviations: APACHE, acute physiology and chronic health evaluation; CRRT, continuous renal replacement therapy; DIC, disseminated intravascular coagulation; GI, gastrointestinal; ICU, intensive care unit; IQR, interquartile range; MOD, multiple organ dysfunction; N/A, not available.

a A total of 29 patients died within 30 days from admission, while 1 patient died after 30 days. Three patients in the fatal group had missing data on 30-day survival time as they died outside of the hospital.

b Analysis using Chi-square test, Fisher's exact test, or Mann-Whitney U test.

c Missing data: n = 2.

d A total of 23 patients in non-fatal group and 10 patients in fatal group received dexamethasone, 3 in non-fatal and 1 in fatal groups received prednisolone, 9 in non-fatal and 7 in fatal groups received methylprednisolone, and 3 in non-fatal and 7 in fatal groups received hydrocortisone. Of 58 patients who received steroids therapy, five patients received two types of steroid therapy such as dexamethasone plus hydrocortisone or methylprednisolone and methylprednisolone plus prednisolone or hydrocortisone. The dosages and types of steroids prescribed for each patients were different depends on hospitals because there were no guidelines of using steroids in patients with SFTS.

e Missing data: n = 1.

f Other complications included acute cerebral infarction, acute respiratory distress syndrome, aspiration pneumonia, azotemia, infectious mononucleosis, interstitial pulmonary fibrosis, intracranial hemorrhage, lymphadenitis, metabolic encephalopathy, multiple organ failure, pulmonary hemorrhage, respiratory failure, and toxic hepatitis.

use was the most commonly observed (n = 33, 23.4%), followed by acute kidney injury (n = 12, 8.5%), pneumonia (n = 11, 7.8%), and septic shock/sepsis (n = 11, 7.8%). Another observed complication was disseminated intravascular coagulation (n = 4, 2.8%) (Table 1).

Among the 142 patients, the initial major symptoms of 140 patients were as follows: fever (n = 121, 86.4%), chills (n = 81, 57.9%), myalgia (n = 60, 42.9%), diarrhea (n = 42, 30%), and nausea (n = 38, 27.1%). There were significant differences in the prevalence of fever (P = 0.033), chills (P = 0.005), myalgia (P = 0.008), and altered mental state (P = 0.028) between the non-fatal and fatal groups (Table 1).

Of the 142 patients, 58 were administered steroid therapy, of which 23 (39.7%) patients died. Most patients who underwent steroid therapy were administered 5–30 mg of dexamethasone (33/58 patients) intravenously or 40–1,000 mg of methylprednisolone (16/58 patients). The median total steroid administration period was 4.5 days (IQR 3–7.5 days).

## Steroid group of patients with SFTS

There was no statistical difference in age or sex between the steroid and non-steroid groups (P = .144, P = .889, Table 2). At the time of hospital admission, the median APACHE II score (IQR) differed significantly between the steroid group, 13.0 (9.0–17.0), and the non-steroid group, 10.0 (8.5–14.0), (P = .042). The ICU admission rate was statistically higher in the steroid group (67.2%, 39 patients) than in the non-steroid group (27.7%, 23 patients) (P < .001). In combined treatment, the use of ribavirin, plasmapheresis, and IVIG accounted for 34.5% (n = 20), 25.9% (n = 15), and 43.1% (n = 25) of patients, respectively, in the steroid group, which was statistically higher than those in the non-steroid group (P < .001, P = .014, and P = .032, respectively) (Table 2).

Among 58 patients who received steroid therapy, comparisons of laboratory test results and clinical characteristics before and after treatment for 41 patients (surviving patients, n = 27 [65.9%]; fatal patients, n = 14 [34.1%]) who underwent laboratory testing both pre- and post-steroid treatment are shown in Table 3.

**Table 2. General, Clinical Characteristics, and Laboratory Results of Patients with SFTS in the Steroid and Non-Steroid Groups (2013–2017).**

| Characteristics | Non-Steroid n = 84 | | Steroid n = 58 | | Total N = 142 | | P Value[a] |
|---|---|---|---|---|---|---|---|
| Age, years (median, IQR) | 67 | (61.0–73.8) | 71.5 | (62.5–77.0) | 68.5 | (61.0–75.3) | .144 |
| Sex, Male | 43 | (51.2) | 29 | (50.0) | 72 | (50.7) | .889 |
| Comorbidity, total[b] | 50 | (59.5) | 38 | (65.5) | 88 | (62.0) | .470 |
| Vital sign at first clinic visit | | | | | | | |
| Body temperature (˚C) (median, IQR) | 37.9 | (37.1–38.5) | 38.2 | (37.2–38.8) | 38.0 | (37.1–38.6) | .184 |
| SBP (mmHg) (median, IQR) | 116.0 | (107.3–130.0) | 110.0 | (100.0–125.3) | 111.5 | (100.0–128.3) | .033 |
| DBP (mmHg) (median, IQR) | 70.0 | (60.0–77.8) | 65.0 | (60.0–77.3) | 70.0 | (60.0–77.3) | .224 |
| Heart rate (/min) (median, IQR) | 79.0 | (69.0–90.0) | 84.0 | (74.3–92.3) | 80.5 | (70.0–90.0) | .100 |
| Respiration rate (/min) (median, IQR) | 20.0 | (18.0–20.0) | 20.0 | (20.0–22.0) | 20.0 | (20.0–22.0) | .028 |
| Symptom onset to admission (median days, IQR) | 5.0 | (3.0–6.0) | 5.0 | (3.0–5.3) | 5.0 | (3.0–7.0) | .908 |
| Initial APACHE II score (median, IQR) | 10.0 | (8.5–14.0) | 13.0 | (9.0–17.0) | 11 | (9.0–16.0) | .042 |
| Initial clinical manifestation[c] | | | | | | | |
| Fever | 76 | (91.6) | 45 | (78.9) | 121 | (86.4) | .018 |
| Chills | 54 | (65.1) | 27 | (47.4) | 81 | (57.9) | .033 |
| Myalgia | 42 | (50.6) | 18 | (31.6) | 60 | (42.9) | .038 |
| Gastrointestinal[d] | 55 | (66.3) | 39 | (68.4) | 94 | (67.1) | .709 |
| Central nervous system[e] | 40 | (48.2) | 29 | (50.9) | 69 | (49.3) | .887 |
| Glasgow Coma Scale (median, IQR) | 15 | (15–15) | 15 | (13–15) | 15 | (15–15) | .008 |
| Initial laboratory findings | | | | | | | |
| Leukopenia (<4,000/mm$^3$) | 76 | (90.5) | 50 | (86.2) | 126 | (88.7) | .429 |
| Neutropenia (ANC <1,500/mm$^3$) | 52 | (68.4) | 39 | (67.2) | 91 | (67.9) | .885 |
| Lymphopenia (ALC <1500/mm$^3$) | 74 | (96.1) | 57 | (98.3) | 131 | (97.0) | .634 |
| Thrombocytopenia, mild (<150 × 10$^3$/mm$^3$) | 81 | (96.4) | 55 | (94.8) | 136 | (95.8) | .688 |
| Anemia (<11 g/dL) | 7 | (8.3) | 4 | (6.9) | 11 | (7.7) | 1.000 |
| Hypoalbuminemia (<3.5 g/dL) | 28 | (38.4) | 27 | (48.2) | 55 | (42.6) | .262 |
| Elevated ALP (>120 IU/L) | 18 | (29.0) | 16 | (31.4) | 34 | (30.1) | .787 |
| Elevated AST (>40 IU/L) | 69 | (84.1) | 46 | (80.7) | 115 | (82.7) | .597 |
| Elevated AST, high (>200 IU/L) | 29 | (35.4) | 22 | (38.6) | 51 | (36.7) | .697 |
| Elevated ALT, (>40 IU/L) | 49 | (59.8) | 36 | (63.2) | 85 | (61.2) | .686 |
| Elevated ALT, high (>200 IU/L) | 11 | (13.4) | 6 | (10.5) | 17 | (12.2) | .609 |
| PT prolongation (INR >1.3) | 3 | (4.2) | 4 | (8.2) | 7 | (5.8) | .442 |
| aPTT prolongation (>40 s) | 46 | (64.8) | 29 | (59.2) | 75 | (62.5) | .533 |
| Elevated CK (>300 IU/L) | 41 | (66.1) | 31 | (81.6) | 72 | (72.0) | .095 |
| Elevated LDH (>300 IU/L) | 47 | (82.5) | 36 | (90.0) | 83 | (85.6) | .298 |
| Elevated CRP (>3 mg/dL) | 9 | (11.8) | 3 | (5.9) | 12 | (9.4) | .359 |
| Prior antibiotic treatment | 28 | (34.6) | 27 | (46.6) | 55 | (39.6) | .154 |
| CRRT/hemodialysis | 5 | (6.0) | 11 | (19.0) | 16 | (11.3) | .016 |
| ICU admission during hospitalization | 23 | (27.7) | 39 | (67.2) | 62 | (44.0) | < .001 |
| Combined treatment | | | | | | | |
| Ribavirin | 14 | (16.7) | 20 | (34.5) | 34 | (23.9) | .014 |
| Plasmapheresis | 10 | (11.9) | 15 | (25.9) | 25 | (17.6) | .032 |
| IVIG | 4 | (4.8) | 25 | (43.1) | 29 | (20.4) | < .001 |
| Steroid dose (median, IQR) [f] | | | | | | | |
| Dexamethasone (mg/day) | N/A | | 15.0 | (10.0–18.0) | 15.0 | (10.0–18.0) | N/A |
| Prednisolone (mg/day) | N/A | | 15.0 | (10.0–20.0) | 15.0 | (10.0–20.0) | N/A |
| Methylprednisolone (mg/day) | N/A | | 68.8 | (56.3–417.5) | 68.8 | (56.3–417.5) | N/A |
| Hydrocortisone (mg/day) | N/A | | 200.0 | (200.0–300.0) | 200.0 | (200.0–300.0) | N/A |

(*Continued*)

**Table 2.** (Continued)

| Characteristics | Non-Steroid | | Steroid | | Total | | P Value[a] |
|---|---|---|---|---|---|---|---|
| | n = 84 | | n = 58 | | N = 142 | | |
| Fatality | 10 | (11.9) | 23 | (39.7) | 33 | (23.2) | < .001 |

Data are presented as no. (%) unless otherwise indicated.

Abbreviations: ALC, absolute lymphocyte count; ALP, alkaline phosphatase; ALT, alanine aminotransferase; ANC, absolute neutrophil count; APACHE, acute physiology and chronic health evaluation; aPTT, activated partial thromboplastin time; AST, aspartate aminotransferase; CK, creatinine kinase; CRP, C-reactive protein; CRRT, continuous renal replacement therapy; DIC, disseminated intravascular coagulation; DBP, diastolic blood pressure; G-I, gastro-intestinal; HLH, hemophagocytic lymphohistiocytosis; ICU, intensive care unit; INR, international normalized ratio; IQR, interquartile range; IVIG, intravenous immunoglobulin; LDH, lactate dehydrogenase; MOD, multiple organ dysfunction; N/A, not available; PT, prothrombin time; SBP, systemic blood pressure.

a Analysis using Chi-square test, Fisher's exact test or Mann-Whitney U test.

b More details of the information are presented in S2 Table.

c Missing data = 2 (non-steroid, n = 1; steroid, n = 1).

d Gastrointestinal symptoms included anorexia, nausea, vomiting, diarrhea, abdominal pain, and abdominal tenderness. More details of the information are presented in S2 Table.

e Central nervous system symptoms included headache, dizziness, neck stiffness, and altered mentation. More details of the information are presented in S2 Table.

f A total of 33 patients received dexamethasone, 4 received prednisolone, 16 received methylprednisolone, and 10 received hydrocortisone. Five patients received two types of steroid therapy such as dexamethasone plus hydrocortisone or methylprednisolone and methylprednisolone plus prednisolone or hydrocortisone.

White blood cell count within 48 hours post-steroid treatment increased to 3,200/μL from 1,700/μL before treatment ($P < .001$). Platelet count also increased to $60.0 \times 10^3/mm^3$ compared to a pre-treatment value of $51.5 \times 10^3/mm^3$ ($P = .033$). Aspartate aminotransferase (AST)/alanine aminotransferase (ALT) levels increased to 416/129 U/L post-treatment from 304/99 U/L pre-treatment ($P = .024/P = .026$), and blood urea nitrogen (BUN) levels increased to 22.8 mg/dL post-treatment from 16.1 mg/dL pre-treatment ($P = .021$). Body temperature ($P < .001$), albumin ($P = .005$), hemoglobin ($P < .001$), and hematocrit ($P < .001$) levels were significantly lower post-treatment compared to pre-treatment values (Table 3).

### Risk factors associated with 30-day mortality in patients with SFTS

The results of univariate analysis indicated that age, arrhythmia, septic shock/sepsis, initial APACHE II score, ICU admission, initial altered mental state, mechanical ventilator use, and CRRT/hemodialysis treatment affected 30-day mortality rates. In the multivariate analysis, age (adjusted hazard ratio [aHR] 1.10, 95% confidence interval [CI] 1.04–1.17), arrhythmia (aHR 4.61, 95% CI 1.42–14.94), septic shock/sepsis (aHR 4.52, 95% CI 1.33–15.38), initial APACHE II score (aHR 1.08, 95% CI 1.01–1.15), ICU admission (aHR 41.90, 95% CI 4.51–389.16), mechanical ventilator use (aHR 2.87, 95% CI 1.01–8.13), and CRRT/hemodialysis treatment (aHR 4.34, 95% CI 1.36–13.89) were statistically significant (Table 4).

A Cox proportional multivariable regression analysis was conducted to examine the effects of various therapies on 30-day mortality in patients with SFTS. Among therapies such as prior antibiotic treatment, ribavirin, steroid therapy, IVIG, and plasmapheresis, the use of steroid therapy was associated with an increased risk of 30-day mortality (aHR 3.45, 95% CI 1.31–9.11, $P = .012$) (Table 5).

### Survival time estimation based on steroid use before and after propensity score matching

Patient clinical characteristics before and after cohort matching are shown in Table 6. Comparisons of survival according to the use of steroid therapy revealed a significant difference in

**Table 3. Comparison of Clinical and Laboratory Parameters in Patients with SFTS Within 48 Hours Before and After Steroid Treatment.**

| Variables | Within 48 h before treatment | | | Within 48 h after treatment | | | (n = 41) |
|---|---|---|---|---|---|---|---|
| | n | Median | (IQR) | n | Median | (IQR) | P Value[a] |
| GCS score | 39 | 14.0 | (11.0–15.0) | 39 | 12.0 | (8.0–15.0) | .002 |
| BT (˚C) | 30 | 38.3 | (37.6–38.7) | 39 | 37.0 | (36.4–37.7) | < .001 |
| WBC (/µL) | 40 | 1700.0 | (1140.0–2837.5) | 41 | 3200.0 | (2500.0–5150.0) | < .001 |
| Neutrophil (%) | 40 | 59.9 | (47.1–69.7) | 38 | 59.6 | (43.5–69.3) | .934 |
| ANC | 40 | 1047.9 | (615.1–1574.7) | 38 | 2010.3 | (1110.9–3236.1) | < .001 |
| Lymphocyte (%) | 40 | 27.3 | (17.5–35.4) | 38 | 28.3 | (17.0–37.8) | .695 |
| ALC | 40 | 409.1 | (291.7–826.2) | 38 | 794.8 | (520.2–1467.2) | .003 |
| Hgb (g/dL) | 40 | 12.7 | (11.9–14.5) | 41 | 12.0 | (10.2–13.3) | < .001 |
| Hct (%) | 40 | 36.5 | (34.3–40.4) | 41 | 34.4 | (30.2–38.2) | < .001 |
| Platelet (10^3/µL) | 40 | 51.5 | (32.3–61.8) | 41 | 60.0 | (38.0–83.5) | .033 |
| Protein (g/dL) | 34 | 5.5 | (4.8–6.1) | 32 | 5.4 | (4.6–6.1) | .553 |
| Albumin (g/dL) | 36 | 3.0 | (2.6–3.4) | 34 | 2.7 | (2.4–2.9) | .005 |
| T-bilirubin (mg/dL) | 38 | 0.4 | (0.3–0.6) | 36 | 0.6 | (0.4–0.9) | .013 |
| ALP (IU/L) | 37 | 85.0 | (57.0–163.5) | 35 | 101.0 | (57.0–160.0) | .464 |
| r-GTP (U/L) | 20 | 60.5 | (23.8–139.8) | 18 | 142.0 | (74.3–295.5) | .209 |
| AST (U/L) | 39 | 304.0 | (173.0–487.0) | 40 | 416.0 | (212.5–860.5) | .024 |
| ALT (U/L) | 39 | 99.0 | (57.0–135.0) | 40 | 129.0 | (72.3–229.3) | .026 |
| BUN (mg/dL) | 39 | 16.1 | (11.7–24.1) | 39 | 22.8 | (14.3–29.9) | .021 |
| Serum Cr (mg/dL) | 38 | 0.8 | (0.7–1.2) | 38 | 0.9 | (0.7–1.4) | .483 |
| Amylase | 21 | 136.0 | (86.0–211.5) | 7 | 231.0 | (184.0–351.0) | .345 |
| CK (IU/L) | 26 | 2134.0 | (934.3–2574.3) | 24 | 1470.5 | (866.3–2870.8) | .407 |
| CK-MB (IU/L) | 16 | 5.2 | (3.1–28.7) | 10 | 16.1 | (5.8–24.3) | .500 |
| LDH (IU/L) | 28 | 1100.5 | (823.0–2520.8) | 34 | 2024.0 | (1125.8–3695.0) | .014 |
| CRP (mg/dL) | 32 | 0.5 | (0.1–1.2) | 20 | 0.5 | (0.0–1.7) | .727 |
| Ferritin (ng/mL) | 11 | 3399.4 | (3180.4–16500.0) | 11 | 7325.5 | (2802.3–9504.0) | .593 |
| Serum Na (mEq/L) | 39 | 138.0 | (136.0–140.0) | 35 | 139.0 | (135.0–140.0) | .027 |
| Serum K (mEq/L) | 39 | 3.9 | (3.6–4.2) | 35 | 4.1 | (3.6–4.5) | .224 |
| INR | 28 | 1.1 | (1.0–1.2) | 29 | 1.0 | (1.0–1.1) | .217 |
| aPTT (s) | 28 | 52.3 | (44.2–70.9) | 28 | 44.6 | (34.5–64.9) | .958 |
| Fibrinogen (mg/dL) | 12 | 220.4 | (169.8–248.2) | 15 | 174.7 | (142.2–204.0) | .063 |

Abbreviations: ALC, absolute lymphocyte count; ALP, alkaline phosphatase; ALT, alanine aminotransferase; ANC, absolute neutrophil count; aPTT, activated partial thromboplastin time; AST, aspartate aminotransferase; BT, body temperature; CK, creatinine kinase; CK-MB, creatine kinase-myocardial band; Cr, creatinine; CRP, c-reactive protein; GCS, Glasgow Coma Scale; Hct, hematocrit; Hgb, hemoglobin; INR, international normalized ratio; K, potassium; LDH, lactate dehydrogenase; Na, sodium; r-GTP, gamma-glutamic transpeptidase; WBC, white blood cell.
a Analysis using Wilcoxon signed-rank test.

mean survival time between the steroid and non-steroid groups (20.8 vs. 27.5 days, $P < .001$, log-rank test) before matching. After matching, the difference in mean survival times between the steroid and non-steroid groups remained significant (21.3 vs. 27.3 days, $P = .002$, log-rank test) (Fig 2A). Comparison of 30-day mortality between the steroid and non-steroid groups in 135 of 142 patients with a confirmed survival time showed a significant difference in survival rates between steroid and non-steroid groups (60.3% [35/58] vs. 90.9% [70/77], $P < .001$, chi-squared test) before propensity score matching. After propensity score matching, the difference remained significant, with a higher survival rate of 90.4% (47/52) in the non-steroid group compared to 62.5% (35/56) in the steroid group ($P = .001$, Table 6).

**Table 4. Risk Factors Associated with 30-day Mortality in Patients with SFTS.**

| Variable | Univariate Model | | | Multivariate Model[a] | | |
|---|---|---|---|---|---|---|
| | HR | (95% CI) | P Value | aHR | (95% CI) | P Value |
| Age | 1.08 | (1.03–1.12) | .001 | 1.10 | (1.04–1.17) | .001 |
| Arrhythmia | 6.23 | (2.53–15.38) | < .001 | 4.61 | (1.42–14.94) | .011 |
| Septic shock/Sepsis | 5.32 | (2.36–11.99) | < .001 | 4.52 | (1.33–15.38) | .016 |
| Initial APACHE II score | 1.10 | (1.06–1.14) | < .001 | 1.08 | (1.01–1.15) | .033 |
| ICU admission | 13.27 | (4.02–43.80) | < .001 | 41.90 | (4.51–389.16) | .001 |
| Initial altered mentation | 2.62 | (1.21–5.64) | .014 | - | | |
| Initial PT prolongation (INR >1.3) | 2.95 | (0.88–9.90) | .080 | - | | |
| Mechanical ventilator | 8.21 | (3.83–17.61) | < .001 | 2.87 | (1.01–8.13) | .047 |
| CRRT/Hemodialysis | 5.74 | (2.72–12.11) | < .001 | 4.34 | (1.36–13.89) | .013 |

Abbreviations: APACHE, acute physiology and chronic health evaluation; aHR, adjusted hazard ratio; CI, confidence interval; CRRT, continuous renal replacement therapy; DIC, disseminated intravascular coagulation; HR, hazard ratio; ICU, intensive care unit; INR, international normalized ratio; PT, prothrombin time.
a The variables included in the multivariate model were age, arrhythmia, septic shock/sepsis, initial APACHE II score, ICU admission, Initial altered mentation, initial PT prolongation, mechanical ventilator and CRRT/hemodialysis.

In the analysis prior to propensity score matching, the early steroid group, which was administered steroids within 5 days of symptom onset, had a 30-day survival time of 18.42 days, which was shorter than that of the late steroid group, which was administered steroid therapy after 5 days of symptom onset, although this difference was not statistically significant ($P = .477$). Differences in survival time between the non-steroid and the early and late steroid groups were statistically significant ($P < .001$ and $P = .001$, respectively) (Fig 2B). Analysis between the non-steroid, early, and late steroid groups after matching showed significant differences in survival times between the non-steroid and early steroid groups ($P = .002$), with survival times of 18.4, 22.4, and 27.3 days for early, late, and non-steroid groups, respectively ($P = .005$) (Fig 2C).

In the analysis prior to matching, in patients with severe disease (an initial APACHE II score $\geq$14), the 30-day survival time did not differ significantly at 17.5 and 21.4 days for the steroid and non-steroid groups, respectively ($P = .307$) (Fig 3A). After matching, there was also no statistically significant difference between the groups (17.5 vs. 22.7 days, $P = .184$) (Fig 3B). However, a significant difference in mortality following steroid administration was observed for patients with APACHE II scores <14. Prior to matching, the 30-day survival time was significantly lower in the steroid group than in the non-steroid group (24.3 [20.55–27.98] vs. 29.4 [95% CI 28.19–30.58] days, $P = .003$) (Fig 3A). After matching, the difference in

**Table 5. Treatment Effects on 30-day Mortality in Patients with SFTS.**

| Variables | Univariate Analysis | | | Multivariate Analysis[a] | | |
|---|---|---|---|---|---|---|
| | HR | (95% CI) | P Value | aHR | (95% CI) | P Value |
| Prior antibiotic treatment | 1.55 | (0.76–3.16) | .234 | - | | |
| Ribavirin | 1.61 | (0.75–3.45) | .217 | - | | |
| Steroids | 4.57 | (1.96–10.66) | < .001 | 3.45 | (1.31–9.11) | .012 |
| IVIG | 1.61 | (0.74–3.51) | .235 | - | | |
| Plasmapheresis | 2.19 | (1.03–4.68) | .043 | - | | |

Abbreviations: aHR, adjusted hazard ratio; CI, confidence interval; HR, hazard ratio; IVIG, intravenous immunoglobulin
a The variables included in the multivariate model were prior antibiotic treatment, ribavirin, steroids, IVIG, plasmapheresis, and adjusted variables such as age, sex, initial APACHE II score, and symptom onset to admission within 7 days.

**Table 6. Characteristics of the SFTS Cohort Before and After Propensity Score Matching.**

| Variables | Unmatched | | | | | Matched | | | | |
|---|---|---|---|---|---|---|---|---|---|---|
| | Non-Steroid | | Steroid | | P Value[a] | Non-Steroid | | Steroid | | P Value[a] |
| | (n = 84) | | (n = 58) | | | (n = 56) | | (n = 56) | | |
| Age[b] (median, IQR) | 67.0 | (61.0–73.8) | 71.5 | (62.5–77.0) | .144 | 69.0 | (65.0–77.0) | 72.5 | (63.0–77.0) | .775 |
| Female, sex[b] | 41 | (48.8) | 29 | (50.0) | .889 | 28 | (50.0) | 29 | (51.8) | .850 |
| Underlying comorbidities[b] | 50 | (59.5) | 38 | (65.5) | .470 | 37 | (66.1) | 37 | (66.1) | 1.000 |
| Initial respiration rate[b,c] (median, IQR) | 20.0 | (18.0–20.0) | 20.0 | (20.0–22.0) | .028 | 20.0 | (20.0–20.0) | 20.0 | (20.0–22.0) | .254 |
| Initial APACHE II score[b,c] (median, IQR) | 10.0 | (8.5–14.0) | 13.0 | (9.0–17.0) | .042 | 11.0 | (9.3–15.0) | 13.0 | (9.0–17.0) | .186 |
| Initial ICU admission[b,c] | 11 | (13.4) | 14 | (24.1) | .103 | 11 | (19.6) | 14 | (25.0) | .496 |
| Initial altered mentation[b,c] | 15 | (19.2) | 14 | (25.0) | .424 | 12 | (21.4) | 14 | (25.0) | .654 |
| Survival | 70 | (90.9) | 35 | (60.3) | < .001 | 47 | (90.4) | 35 | (62.5) | < .001 |

Data are presented as no. (%) unless otherwise indicated.

Abbreviations: APACHE, acute physiology and chronic health evaluation; ICU, intensive care unit; IQR, interquartile range.

a Analysis using Chi-square test or Mann-Whitney U test.

b Matching variables.

c 'Initial ICU admission' means an intensive care unit admission at the beginning of hospitalization. 'Initial altered mentation' means altered mentation between onset and admission. Missing data for variables such as initial APACHE II score (n = unmatched non-steroid group 15, unmatched steroid group 1), initial ICU admission (unmatched non-steroid 2), initial altered mentation (unmatched non-steroid group 6, unmatched steroid group 2), initial GCS score (unmatched non-steroid group 12, unmatched steroid group 6, matched non-steroid group 4, matched steroid group 4), and complications (unmatched non-steroid group 1).

survival time was statistically significant (24.9 [21.21–28.53] vs. 29.2 [27.70–30.73] days, *P* = .022) in patients with mild-to-moderate disease (APACHE II scores <14) (Fig 3B).

## Complications in patients with SFTS Using steroid therapy

After propensity score matching, the difference in the incidence of complications between the steroid and non-steroid groups was statistically significant (*P* < .001), with 62.5% (35/56) and 28.6% (16/56) of patients experiencing complications in the steroid group and non-steroid groups, respectively (Table 7). More patients required mechanical ventilation in the steroid group (33.9%, 19/56) than in the non-steroid group (12.5%, *P* = .007), and the incidence of pneumonia was higher in the former than in the latter (14.3% vs. 1.8%, *P* = .032).

## Discussion

SFTS is an emerging infectious disease that was first identified in 2011 [5]. This tick-borne infection has been mainly reported in Korea, China, and Japan [9]. The number of patients with SFTS has increased annually in Korea [6]. However, few studies have analyzed the efficacy of SFTS treatment. In patients with Crimean–Congo hemorrhagic fever, which is similar to SFTS in terms of being a zoonotic disease transmitted by ticks and characterized by fever and hemorrhage, the administration of high-dose methylprednisolone increases platelet counts and reduces the requirement for blood products [17]. Moreover, three case reports from Japan demonstrated the efficacy of short-term glucocorticoid therapy for SFTS accompanied by encephalopathy [11]. However, in the current study, 48 hours before and after steroid treatment, the patients GCS score reduced, which is contradictory to the results of the previous study. Although this study was limited by the use of various steroids and doses, it confirmed the effect of steroids on SFTS patients.

In early-stage SFTS infection, the function and differentiation of T follicular helper cells are disrupted by impaired antigen presentation due to monocyte and dendritic cell apoptosis,

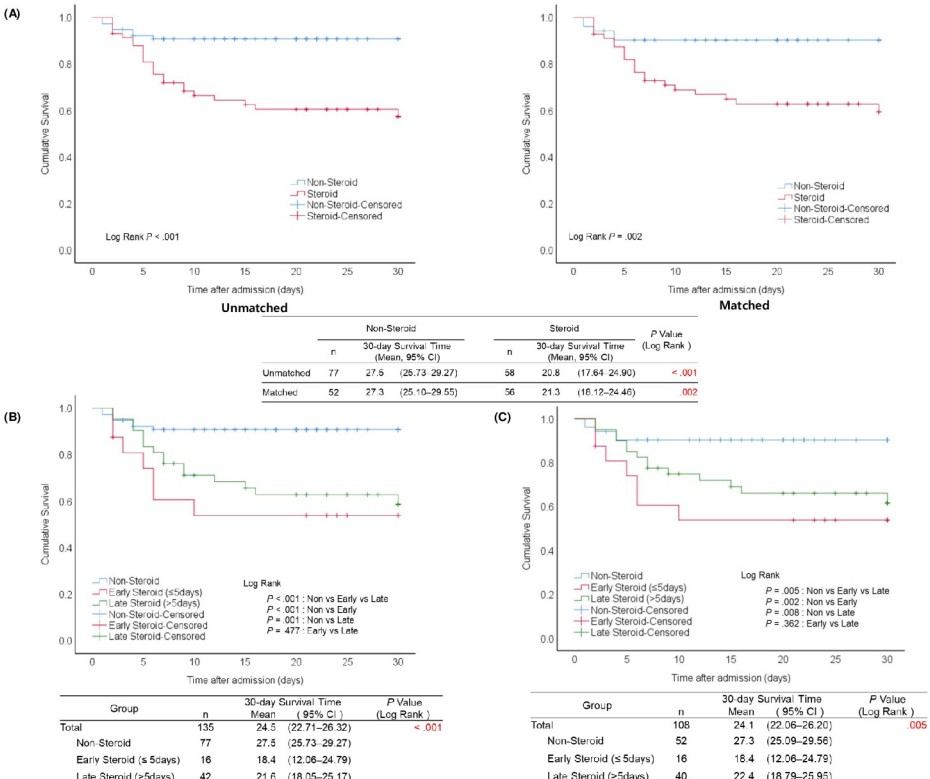

**Fig 2. Survival analysis according to the timing of steroid therapy in patients with severe fever with thrombocytopenia syndrome (SFTS).** (A) Kaplan–Meier analysis of 30-day survival after admission between patients with severe fever with thrombocytopenia syndrome (SFTS) with and without steroid therapy before and after propensity score matching[a]. (B) Kaplan–Meier analysis of 30-day survival after admission among patients with severe fever with thrombocytopenia syndrome (SFTS) with early, late, and non-steroid treatment, before propensity score matching. (C) Kaplan–Meier analysis of 30-day survival after admission among patients with severe fever with thrombocytopenia syndrome (SFTS) with early, late, and non-steroid treatment, after propensity score matching. a. There were 7 missing patients in the unmatched dataset and 4 in the matched dataset due to missing data on 30-day survival time.

which ultimately leads to failure of virus-specific humoral response [18]. Furthermore, previous studies have reported that steroid administration resulted in enhanced T and B cell (CD4[+] T cells, CD8[+] T cells, and CD19[+] B cells) apoptosis, suggesting a potentially negative effect on immune function following steroid administration [19,20]. This research constitutes the background to our study.

In a simple comparative analysis of treatment methods, steroid therapy was administered to 40.8% (58/142) of patients, of which a higher proportion died after having received steroid therapy [69.7% of those who died and 32.1% of survivors received steroid therapy ($P < .001$)]. However, this simple comparison of mortality according to treatment was not useful for our analysis of therapeutic effects due to possible biases; for example, patients with severe disease were more likely to be administered steroid therapy or ribavirin. Therefore, it was necessary to match and analyze each treatment group based on severity and risk factors. With regard to the risk factors for mortality, the APACHE II score and the frequency of symptoms such as febrile sensation and chills differed. We confirmed that symptoms were subjective data; as the APACHE II score increased, the patient had lesser time to express their symptoms due to loss of consciousness. For the risk factor as confounders of treatment effect, we did not only

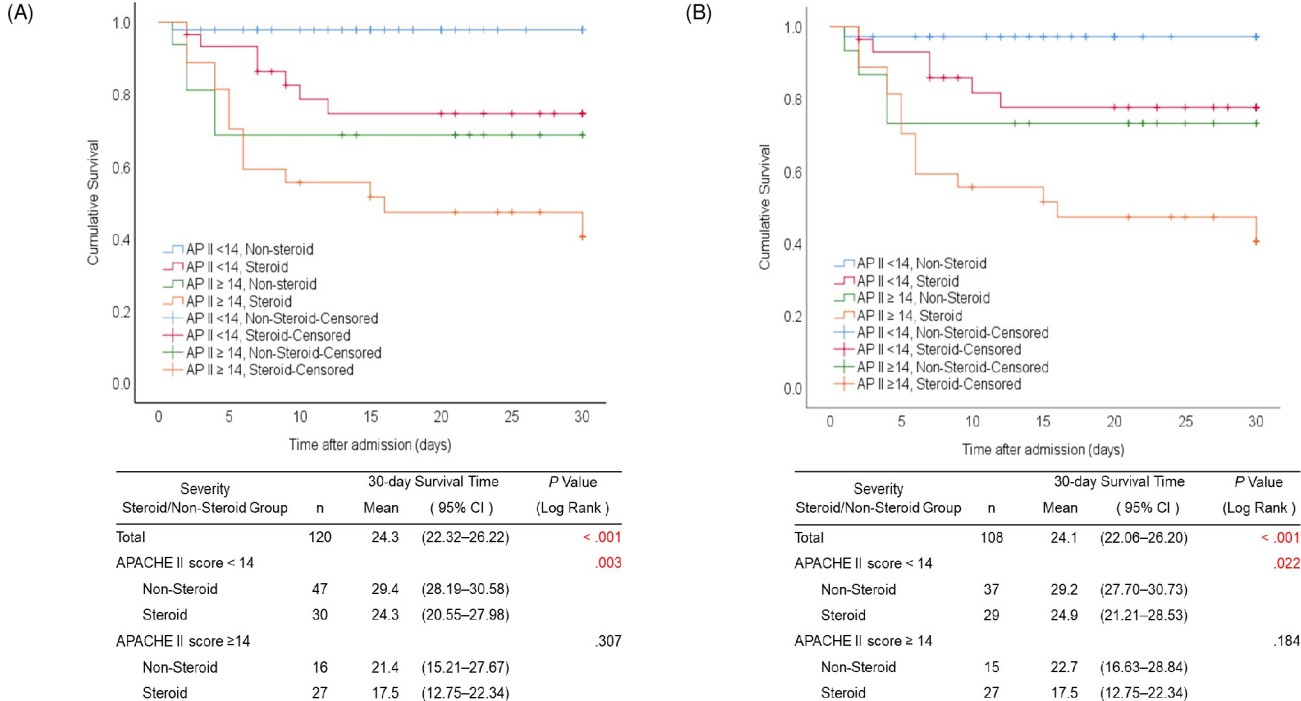

**Fig 3. Survival analysis according to Acute Physiology and Chronic Health Evaluation (APACHE) II scores among patients with severe fever with thrombocytopenia syndrome (SFTS).** (A) Kaplan–Meier 30-day survival analysis after admission of patients with severe fever with thrombocytopenia syndrome (SFTS) with and without steroid therapy according to SFTS severity before propensity score matching[a]. (B) Kaplan–Meier 30-day survival analysis of patients with severe fever with thrombocytopenia syndrome (SFTS) with and without steroid therapy according to SFTS severity after propensity score matching[b]. a. Twenty-two patients had missing data on the initial APACHE II score or 30-day survival time. b. Four patients had missing data on 30-day survival time.

include the patient's subjective symptoms but also the objective data such as laboratory results. Thus, we employed propensity score matching for a more accurate comparison between the two treatment groups.

In this study, the survival time in the steroid group was lower than in the non-steroid group both before and after propensity score matching, especially between the early (symptom onset within 5 days) and non-steroid groups. Analysis of 30-day survival after matching in the steroid and non-steroid groups according to SFTS severity revealed a significantly shorter survival time in the steroid group than in the non-steroid group for APACHE II scores <14. In addition, univariate and multivariate analyses of 30-day survival showed increased mortality in the steroid group (aHR: 3.31, $P$ = .016). Therefore, steroid administration may be more harmful than beneficial in patients with SFTS. The early use of hydrocortisone within 1 week of symptom onset in patients with severe acute respiratory syndrome (SARS) may cause higher plasma viral loads in the second and third weeks of symptom onset [21]. This suggests that there is a need for additional prospective studies concerning the effects of steroid administration on SFTSV kinetics while identifying the potential side effects of steroid therapy such as immunosuppressive effects, bacterial/fungal superinfection, hyperglycemia, electrolyte imbalance, and psychosis [22,23].

A retrospective analysis of Japanese patients with SFTS from 2013 to 2014 reported hemophagocytosis in 15 of 18 patients who underwent bone marrow examination. It also showed that steroid administration should be considered when treating hemophagocytosis [7]. However, 10% of patients from the same study experienced fungal infections such as invasive

**Table 7. Comparison of Complication Frequencies Between the Steroid and Non-Steroid Groups in Patients with SFTS after Propensity Score Matching.**

| Variables | Non-Steroid (n = 56) | | Steroid (n = 56) | | P Value[a] |
|---|---|---|---|---|---|
| | n(%) | | n(%) | | |
| Complication, total | 16 | (28.6) | 35 | (62.5) | < .001 |
| Meningoencephalitis | 0 | (0.0) | 5 | (8.9) | N/A |
| Mechanical ventilation | 7 | (12.5) | 19 | (33.9) | .007 |
| Arrhythmia | 1 | (1.8) | 6 | (10.7) | .113 |
| Pneumonia | 1 | (1.8) | 8 | (14.3) | .032 |
| Seizure | 0 | (0.0) | 2 | (3.6) | .495 |
| Rhabdomyolysis | 2 | (3.6) | 0 | (0.0) | N/A |
| MOD | 0 | (0.0) | 1 | (1.8) | N/A |
| Septic shock or sepsis | 4 | (7.1) | 7 | (12.5) | .341 |
| Acute kidney injury | 5 | (8.9) | 5 | (8.9) | 1.000 |
| G-I bleeding | 0 | (0.0) | 1 | (1.8) | N/A |
| HLH | 0 | (0.0) | 3 | (5.4) | N/A |
| DIC | 0 | (0.0) | 3 | (5.4) | N/A |
| Others [b] | 8 | (14.3) | 2 | (3.6) | .047 |

Abbreviation: DIC, disseminated intravascular coagulation; G-I, gastro-intestinal; HLH, hemophagocytic lymphohistiocytosis; MOD, multiple organ dysfunction; N/A, not available.

a Analysis using Chi-square test or Fisher's exact test.

b Other complications included acute cerebral infarction, acute respiratory distress syndrome, aspiration pneumonia, azotaemia, infectious mononucleosis, intracranial haemorrhage, metabolic encephalopathy, respiratory failure, and toxic hepatitis.

aspergillosis, and four cases of SFTS accompanied by invasive pulmonary aspergillosis were reported in China [2]. In our study we observed an increase in various complications in those who received steroid therapy after propensity score matching. Although it was uncertain when those complications related to steroids occurred due to retrospective nature of this study, the results of our analysis on the effects of steroid therapy suggest that this therapy may increase mortality rather than provide benefits.

This study had some limitations. This hospital-based study may be not easily generalizable. As this is a retrospective study and not a randomized control trial, it was difficult to assess the effects of steroid monotherapy because patients often received other treatments in combination with steroid therapy. In addition, the sample size in each subgroup may not be sufficient to analyze the effects of treatment based on the type of steroid and doses, time of steroid treatment initiation, or APACHE II score. Additional analysis as part of a systematic prospective study is necessary to determine which infectious complications increase after the administration of steroids as well as the reasons for decreased survival after steroid administration.

In conclusion, we observed increased complications after steroid therapy among patients with SFTS. Steroid therapy should be used with caution considering its possible negative effects on survival within 5 days of symptom onset or in patients with an APACHE II score <14. Further prospective studies on determining the role of steroids therapy is essential for reducing mortality of patients with SFTS.

## Supporting information

**S1 Table. Additional Information of General and Clinical Characteristics of Patients with SFTS in the Non-Fatal and Fatal Groups (2013–2017).**
(DOCX)

**S2 Table. Additional Information on the Clinical Characteristics of Patients with SFTS in the Steroid and Non-Steroid Groups (2013–2017).**
(DOCX)

## Acknowledgments

We thank the following people for providing epidemiological data: Song Mi Moon, M.D., Division of Infectious Diseases, Department of Internal Medicine, Hallym University Sacred Heart Hospital, Anyang, Republic of Korea; and Hong Sang Oh, M.D., Division of Infectious Diseases, Department of Internal Medicine, Armed Forces Capital Hospital, Seongnam, Republic of Korea.

## Author Contributions

**Conceptualization:** Sook In Jung, Dong-Min Kim.

**Data curation:** Sook In Jung, Na Ra Yun, Choon-Mee Kim, Dong-Min Kim, Uh Jin Kim, Seong Eun Kim, Jieun Kim, Seong Yeol Ryu, Hyun ah Kim, Jian Hur, Young Keun Kim, Hye Won Jeong, Jung Yeon Heo, Dong Sik Jung, Hyungdon Lee, Kyungmin Huh, Yee Gyung Kwak, Sujin Lee, Seungjin Lim, Sun Hee Lee, Sun Hee Park, Joon-Sup Yeom, Shin-Woo Kim, In-Gyu Bae, Eu Suk Kim, Jun-Won Seo.

**Formal analysis:** Sook In Jung, Na Ra Yun, Choon-Mee Kim, Dong-Min Kim, Mi Ah Han, Uh Jin Kim, Seong Eun Kim, Jieun Kim, Seong Yeol Ryu, Hyun ah Kim, Jian Hur, Young Keun Kim, Hye Won Jeong, Jung Yeon Heo, Dong Sik Jung, Hyungdon Lee, Kyungmin Huh, Yee Gyung Kwak, Sujin Lee, Seungjin Lim, Sun Hee Lee, Sun Hee Park, Joon-Sup Yeom, Shin-Woo Kim, In-Gyu Bae, Juhyung Lee, Eu Suk Kim, Jun-Won Seo.

**Funding acquisition:** Dong-Min Kim.

**Investigation:** Ye Eun Kim.

**Methodology:** Sook In Jung, Dong-Min Kim.

**Project administration:** Dong-Min Kim.

**Resources:** Sook In Jung, Na Ra Yun, Choon-Mee Kim, Dong-Min Kim, Uh Jin Kim, Seong Eun Kim, Jieun Kim, Seong Yeol Ryu, Hyun ah Kim, Jian Hur, Young Keun Kim, Hye Won Jeong, Jung Yeon Heo, Dong Sik Jung, Hyungdon Lee, Kyungmin Huh, Yee Gyung Kwak, Sujin Lee, Seungjin Lim, Sun Hee Lee, Sun Hee Park, Joon-Sup Yeom, Shin-Woo Kim, In-Gyu Bae, Eu Suk Kim, Jun-Won Seo.

**Software:** Ye Eun Kim.

**Supervision:** Sook In Jung, Dong-Min Kim.

**Validation:** Mi Ah Han, Juhyung Lee.

**Visualization:** Ye Eun Kim.

**Writing – original draft:** Ye Eun Kim.

**Writing – review & editing:** Sook In Jung, Ye Eun Kim, Na Ra Yun, Choon-Mee Kim, Dong-Min Kim, Mi Ah Han, Uh Jin Kim, Seong Eun Kim, Jieun Kim, Seong Yeol Ryu, Hyun ah Kim, Jian Hur, Young Keun Kim, Hye Won Jeong, Jung Yeon Heo, Dong Sik Jung, Hyungdon Lee, Kyungmin Huh, Yee Gyung Kwak, Sujin Lee, Seungjin Lim, Sun Hee Lee, Sun Hee Park, Joon-Sup Yeom, Shin-Woo Kim, In-Gyu Bae, Juhyung Lee, Eu Suk Kim, Jun-Won Seo.

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
