## [Decision Letter · Decision Letter 0]

9 Oct 2020

Dear Dr. Kim,

Thank you very much for submitting your manuscript "Effects of Steroid Therapy in Patients with Severe Fever with Thrombocytopenia Syndrome: A Multicenter Clinical Cohort StudyEffects of Steroid Therapy in Patients with Severe Fever with Thrombocytopenia Syndrome: A Multicenter Clinical Cohort Study" for consideration at PLOS Neglected Tropical Diseases. As with all papers reviewed by the journal, your manuscript was reviewed by members of the editorial board and by several independent reviewers. In light of the reviews (below this email), we would like to invite the resubmission of a significantly-revised version that takes into account the reviewers' comments. 

We cannot make any decision about publication until we have seen the revised manuscript and your response to the reviewers' comments. Your revised manuscript is also likely to be sent to reviewers for further evaluation.

Sincerely,

Anita K. McElroy, MD, PhD

Associate Editor

A. Desiree LaBeaud

Deputy Editor

Reviewer's Responses to Questions

**Key Review Criteria Required for Acceptance?**

**Methods**

-Are the objectives of the study clearly articulated with a clear testable hypothesis stated?

-Is the study design appropriate to address the stated objectives?

-Is the population clearly described and appropriate for the hypothesis being tested?

-Is the sample size sufficient to ensure adequate power to address the hypothesis being tested?

-Were correct statistical analysis used to support conclusions?

-Are there concerns about ethical or regulatory requirements being met?

Reviewer #1: I have no major concerns about the methods. I do think that the sample size is somewhat small to use for propensity score matching, but I understand that patient volume does not currently allow greater numbers, and the authors are trying to achieve a comparison of the treatment between similar groups.

Reviewer #2: See Summary and General Comments

Other comments:

Pg 5, line 109: what were the specific inclusion and exclusion criteria? Was hospitalization an inclusion criterion?

Pg 5, line 116: What other pathogens were routinely tested for? 

Pg 6, line 131: It would be useful to list the components of the APACHE II score (possibly in a supplementary table)

Reviewer #3: The objective of this study was to evaluate the effect of steroids in the treatment of patients hospitalized with SFTSV. The study presented was a multicenter retrospective cohort study, which is an appropriate study design to address this clinical question, given the relative infrequency of infection with this virus. The population under study is well-described, and is sufficiently large for the authors to identify statistically significant effects associated with steroid treatment. 

Of critical concern in the interpretation of retrospective cohort studies is the potential for confounding. In this study, a possible alternative explanation for the author’s findings is that the population receiving steroid treatment is different than the untreated population in some way, for instance, in severity of disease, timing after symptom onset, prominence of respiratory symptoms, etc. Propensity matching is an appropriate method to control for some of these differences, however, concerns regarding residual confounding remain.

Additional comments regarding methods:

Line 112 – 115: The authors state that the diagnosis of SFTS is made by a combination of molecular methods (PCR/RTPCR) and serology. The sensitivity and specificity of these tests are presumably quite different; therefore, it would be useful to specify how many study participants were diagnosed via molecular methods v. serology.

Line 130: To assess differences in 30d mortality, the authors state that they used a Cox regression model adjusted for initial APACHE score and hospitalization within seven days of symptom onset. Why were these variables chosen? Table 2 suggests that there is a small (but significant) difference in APACHE scores between those participants who received steroids and those who did not, but the variable entitled ‘Symptom onset to admission’ in Table 2 was the same between the two groups. Does the description in the methods ‘hospitalization within 7 days of symptom onset’ represent a categorical variable derived from the continuous variable ‘Symptom onset to admission’? This should be clearer.

 Line 140: What do ‘Initial altered mental state’ and ‘initial intensive care unit (ICU) admission’ mean? 

Line 213: Were all of the variables used for univariate analysis included in the multivariate analysis in Table 4?

Line 221: What variables were included in the multivariate analysis presented in Table 5? Are these variables the same as those included in the multivariate analysis in Table 4? This should be clearer.

**Results**

-Does the analysis presented match the analysis plan?

-Are the results clearly and completely presented?

-Are the figures (Tables, Images) of sufficient quality for clarity?

Reviewer #1: Tables 1 and 2 are especially long and more appropriate as supplemental tables. Both include information that I am not sure is necessary for the purposes of the paper, such as tick bit and occupation info in table 1 and two classifications each for elevated AST and ALT. Clinical manifestations could perhaps be reduced to the system and the most important symptoms, such as altered mentation and GCS for the central nervous system. 

It is also unclear to me whether ICU admission in these tables refers to any point in the hospital course or if that is describing where they were initially admitted. My presumption is the former, but then this means that 15% of patients who died were never admitted to the ICU.

The information regarding steroid administration is extremely vague, simply giving a range of doses received without any additional information other than median duration. A dose of dexamethasone 5 mg or methylpred 40 mg is quite different from dexamethasone 30 mg or methylpred 1000 mg. Knowing the distribution in doses would help the reader frame this analysis appropriately. The ideal analysis would stratify by steroid dosing in some way, but it is unlikely there are sufficient numbers for this.

Regarding Table 3, the authors should clarify how values were selected if there was more than 1 in that timeframe? For instance, there were no doubt multiple body temperature measurements in the 48 hours before and after treatment. Did they select the highest temperature, an average of the values, or base in on time criteria? Likely some patients had labs performed more than once in that period as well.

Lastly, additional information on when patients were admitted to the ICU and developed complications in relation to timing of steroids would be appreciated. If most patients received steroids prior to ICU admission, that frames this differently than if most patients received steroids after worsening and transfer to the ICU. The authors note that mechanical ventilation was required more often and DIC was more common in the steroid group. However, it is not at all clear whether these complications developed before or after steroid administration. Without this information, it is very easy to imagine (particularly when considering the late steroid group) that steroids were actually given because of these complications.

Reviewer #2: It would be useful to add the total number of patients tested for SFTS during the time period.

Were all eligible patients included? If not, authors need to explain why not.

Table 1: the numbers of evaluable cases of “memory of tick bite” (n=130) and “presence of bite wound” (n=131) should be added to indicate that some were missing. The statistical test for comparing respiration rates should be checked—the difference does not appear significant although p=0.013

Pg 10, line 177: Steroid therapy: did any local hospital-based guidelines exist that may explain why some patients received steroids whereas others didn’t? Did rates of steroid use differ between hospitals? What type of steroid was used for patients who did not receive dexamethasone or methylprednisolone (n=9)? The authors state that 58 patients received steroids but the numbers who received steroids alone (n=19), steroids + ribavirin (n=20), steroids + IVIG (n=25) and steroids + plasmapheresis (n=15) exceeds 58.

Table 2:

GCS in each group appears to be very similar but p=0.008; stats should be checked

Table 3:

It is not clear why the numbers of patients included in the “before vs after” steroid treatment groups are different because the authors stated on pg 12, line 198 that patients who “underwent laboratory testing BOTH pre- and post-steroid treatment are shown”. Therefore, it seems the numbers should be the same in each group.

Pg 19, line 268: there is a typo—it should read “in the NON-STEROID group than in the STEROID group” not vice versa

Table 7:

It is unnecessary to include n for each variable since they are all the same—n can be included as a footnote.

The figures are of sufficient quality for clarity

Reviewer #3: The presented analysis matches the analysis plan. Overall, the results were clearly presented, and support the author’s conclusions. However, there are several inconsistencies and omissions than should be addressed before the paper is published. In particular, it is disconcerting that the analyses presented in Figures 2 and 3 include different numbers of participants than from each other, or from the table outlining the propensity-matched data set in Table 6. Given the relatively small sample size, the most rigorous approach would be to conduct the analyses presented here using the same propensity matched dataset.

Additional comments regarding results:

Line 149: The results state that of 142 patients, 33 died. Were these deaths captured within a certain period of time after hospitalization? Are these in-hospital deaths? Deaths within 30d of admission? Fatal should be better defined here and in Table 1.

Line 173: These symptoms are noted as ‘initial major symptoms’. Does that mean that only those patients who had fever, chills, etc., at the time of presentation are counted as positive? If these symptoms developed later, are patients still counted as positive?

Line 176: Why are fatal cases less likely to have fever, chills, and myalgias than nonfatal cases? The higher incidence of altered mental status may mean that patients ware less likely to report subjective symptoms (chills, myalgias), but the lower rate of fever raises concerns that patients who end up dying are already different at the time of presentation. Are fatal cases further along in their disease course than nonfatal cases? Are they already receiving some intervention that masks fevers, such as steroids or CRRT? This finding raises concerns about underlying confounding between fatal and nonfatal cases that might bias the analysis of steroids and other interventions – if fatal cases were already sicker at the time of presentation, and are started on CRRT or steroids more rapidly, then the apparent reduction in 30d mortality may reflect confounding rather than effects attributable to an intervention.

Table 177: Of the 58 patients who received steroid therapy, 24 died. This information is not currently available in Table 1 or Table 2. It should be in one place or the other, preferably both. Would include a line in Table 1 denoting steroid treatment, and a line in Table 2 denoting mortality.

Line 177 – 180: Information about steroid dosing should be included in Table 2.

Line 180 – 183: This information is included almost verbatim in the next paragraph. It doesn’t need to be repeated.

Line 212: How were these variables selected for univariate analysis? Were other variables considered?

Line 233 – 238: Does this information about 30d survival appear elsewhere in the study?

Table 6: Why did the number of participants go from 56 to 58 after propensity matching? Might be helpful to know why those two participants were excluded.

Figure 2B: Why were 7 patients excluded (n=135 instead of n = 142)? Should include a footnote explaining the exclusions.

Figure 2C: Why does the non-steroid group now have 52 participants instead of 56 after propensity matching? Is this figure legend incorrect, or is Table 6 incorrect?

Figure 3A: Why does this analysis have 120 instead of 135 participants? Again, if patients are being excluded, should create a footnote detailing why.

Line 286: Would be careful with chi-square P values for groups that have fewer than 5 members; it is difficult to conclude that the frequency of DIC was actually significant given small sample size.

Table 7: Would omit columns for ‘N’, since these are the same for all values; doesn’t add anything to table.

**Conclusions**

-Are the conclusions supported by the data presented?

-Are the limitations of analysis clearly described?

-Do the authors discuss how these data can be helpful to advance our understanding of the topic under study?

-Is public health relevance addressed?

Reviewer #1: Pending response to the issues above, the conclusion by the authors may be too strong. In the last paragraph, they state that they observed increased complications after steroid therapy. They may very well have observed this, but this is not clearly demonstrated by the data presented in the paper. They otherwise are reasonably cautious about whether steroids may cause increased mortality in SFTS. Depending on the distribution of steroid doses, I believe the lack of similar steroid dosing may also be a limitation of the analysis and should be commented on.

The Kaplan-Meier curves both potentially strengthen and weaken their arguments. It is noteworthy that the survival time for the late-steroid group is longer than the early-steroid group, and is perhaps the strongest supporting piece of data for steroids increasing mortality. However, it is also notable that the late-steroid group appears to diverge from the non-steroid group around day 5 to 6, but the late-steroid group by definition received steroids after day 5. Thus, this divergence would not be due to steroids and suggests that matching still left the late-steroid group with sicker patients. This is likely another sign of a limitation of the paper.

This analysis is also limited by a fairly small sample size, particularly with regard to the subgroup analyses based on APACHE score.

Reviewer #2: The changes suggested in the Summary and General Comments as well as other specific comments should be incorporated into the discussion where appropriate.

Reviewer #3: The conclusions reached by the authors are broadly supported by the data as presented. The analyses performed on the propensity matched dataset are the most compelling data presented in this study, and support a conclusion that steroid therapy may lead to adverse outcomes, particularly in mild cases of SFTS. Given the importance of propensity matching to this study, it all the more important that the presented analyses are performed on the same matched dataset. The authors discuss the limitations of their study, and mention the potential for confounding by severity, as sicker patients were more likely to receive steroids and ribavirin, but this remains the largest potential weakness of the study. 

Additional comments regarding conclusions:

Line 316: I don’t think the data regarding 30d survival after steroid therapy for patients with mild disease actually appears in the paper. If it is going to be a discussion point, the data should be included.

**Editorial and Data Presentation Modifications?**

Reviewer #1: (No Response)

Reviewer #2: Major revision required to address the above issues.

Reviewer #3: Line 54: Would capitalize ‘phlebovirus’ as the proper name of a viral genus

Line 64: The reported CI for mean survival in patients with mild disease after steroid treatment is reported in the abstract as 24.9 (95% CI 20.21 – 28.53), but in Figure 3B it is reported as 24.9 (95% CI 21.21 – 28.53). Which is correct?

Line 74: Would capitalize ‘phlebovirus’ as the proper name of a viral genus

Line 170: The text states that of the 142 patients, 130 had confirmed data related to a tick bite. Would recommend including this information in Table 1 (would create a footnote, as was done for ‘Initial clinical manifestation’ and ‘Complications, total’.

Line 162: Would present data from non-fatal group first, followed by fatal group, to match formatting of Table 1. 

Line 171: The line ‘Of the 131 patients with a confirmed diagnosis on the basis of physical examination’ is unclear. Does that mean that only 131 patients had a physical exam performed?

Line 187: The line that ends with (P = .144, P = .889) should cite Table 2, as this is the first reference to that table.

Line 190 – 192: Would reorder these therapies in order that they appear in the table.

Line 202: Keep significant units consistent. Most AST/ALT results are reported to the nearest whole number.

Line 202: The P value for AST is reported as 0.023 in the text, but 0.024 in Table 3.

Figure 2A: Should include a column for ‘n’ in this figure (similar to other figures)

Line 307-308: Unclear what this sentence regarding ribavirin contributes to the discussion or to the analysis overall.

**Summary and General Comments**

Reviewer #1: The authors undertook a retrospective analysis using propensity score matching to attempt to achieve a measure of equipoise in comparing steroids to no steroids in the treatment of SFTS given a lack of data comparing this. As with any retrospective analysis, there are limitations that restrict how strong of a conclusion can be drawn. In particular, the range of steroid dosing appears to be quite wide, and there is a suggestion from the KM curves that those who received steroids were still sicker than those who did not. Presumably, from the wording of the conclusion section, the increased complications in the steroid group occurred after steroids were given, but this is not apparent from the results section and should be clarified. Regardless, there is a suggestion of harm with steroids, and clinicians should carefully consider whether to administer them in patients with SFTS until better data is available.

Reviewer #2: The authors evaluated the effects of steroid therapy on Korean patients hospitalized at multiple centers with documented Severe Fever with Thrombocytopenia Syndrome (SFTS), a potentially fatal tick-borne viral infection. To address potential indication bias in this retrospective study, they performed propensity score matching and showed that mortality was significantly higher in steroid-treated patients who were either less severely ill (low APACHE II) or who were treated early (<5 days after symptom onset). Therefore, they recommend caution when using steroids for SFTS.

This is a well conducted study with rigorous statistical analyses and appropriate adjustments for potential biases. There are no other published studies that have addressed the utility and complications of steroid use for SFTS. Therefore, within its limitations as outlined by the authors, this study may be useful for clinicians where the disease is prevalent (Korea, China and Japan).

However, there are major weaknesses that should be addressed:

1) It appears that the rate of SFTS increased from 2013 to 2017 (Figure 1) but it is unclear if this trend may have been associated with increased testing (other possible reasons could have been an epidemic or increased virulence of the virus). The authors should present the total number of patients tested and the proportion of positive tests each year during that time period. 

2) The potential effect of unidentified co-infections: (pg 6, line 148) “no patient had a confirmed co-infection”. What other infections were tested, were they routinely tested and what proportion of patients were tested? This should be added.

3) It is not clear what the distribution of “time to start of steroids” was in patients who survived vs those who died –this should be added to Table 1.

4) How do the authors explain that frequencies of certain clinical variables were lower in the steroid group vs the non-steroid group (Table 2) such as: fever, chills, myalgia and fatigue although the APACHE II score was higher? This should be discussed.

5) The Glasgow Coma Scale was lower after steroid treatment (Table 3) which should be commented on in the discussion in the context of the paper they cite (Nakamura et al) that recommended steroids for SFTS-associated encephalopathy.

6) On pg 14 (Table 4), the authors describe risk factors for 30-day mortality but most of these are later events and complications (eg ICU admission, mechanical ventilator, hemodialysis) and are not contextualized within the relevant literature describing prediction models that they have not cited such as: 

a. He et al. Severe fever with thrombocytopenia syndrome: a systematic review and meta-analysis of epidemiology, clinical signs, routine laboratory diagnosis, risk factors, and outcomes. BMC Infect Dis. 2020; 20: 575

b. Wang et al. A nomogram to predict mortality in patients with severe fever with thrombocytopenia syndrome at the early stage - A multicenter study in China. PLoS Negl Trop Dis. 2019 Nov; 13(11): e0007829

c. Liu et al. Analysis of the laboratory indexes and risk factors in 189 cases of severe fever with thrombocytopenia syndrome. Medicine (Baltimore). 2020 Jan;99(2):e18727.

d. Li et al. Epidemiological and clinical features of laboratory-diagnosed severe fever with thrombocytopenia syndrome in China, 2011–17: a prospective observational study. Lancet Infect Dis. 2018 Oct;18(10):1127-1137.

7) The authors compared the rates of complications in the steroid vs non-steroid groups (Table 7) but they did not adjust for severity and potential indication bias as they did in Table 6. Therefore, some of these complications may have been related to severe disease and not steroids. Propensity score matching should be added and the discussion should be modified.

Reviewer #3: Overall, this study presents novel findings that may inform treatment for a significant emerging pathogen, and is thus of importance from both a clinical and public health standpoint. The primary strength of this study is novelty, and the incorporation of a number of SFTS patients from several centers over multiple years potentially makes it more generalizable. However, the number of patients included in this study is still relatively small, and with any retrospective cohort study, it is difficult to identify and control for all potential confounding variables. The possibility that patients receiving steroids are more sick, or at a different point in their illness, leading to both poorer outcomes and making it more likely that they will receive steroids, is the biggest potential source of error. Still, the novelty and potential clinical impact of this study are of interest, and warrant publication with minor revisions.

PLOS authors have the option to publish the peer review history of their article (what does this mean?). If published, this will include your full peer review and any attached files.

Reviewer #1: No

Reviewer #2: Yes: Ian C. Michelow, MD, MMed, DTM&H

Reviewer #3: No
---

## [Decision Letter · Decision Letter 1]

28 Dec 2020

Dear Dr. Kim,

Thank you very much for submitting your manuscript "Effects of Steroid Therapy in Patients with Severe Fever with Thrombocytopenia Syndrome: A Multicenter Clinical Cohort Study" for consideration at PLOS Neglected Tropical Diseases. As with all papers reviewed by the journal, your manuscript was reviewed by members of the editorial board and by several independent reviewers. The reviewers appreciated the attention to an important topic. Based on the reviews, we are likely to accept this manuscript for publication, providing that you modify the manuscript according to the review recommendations. 

Sincerely,

Anita K. McElroy, MD, PhD

Associate Editor

A. Desiree LaBeaud

Deputy Editor

Reviewer's Responses to Questions

**Key Review Criteria Required for Acceptance?**

**Methods**

-Are the objectives of the study clearly articulated with a clear testable hypothesis stated?

-Is the study design appropriate to address the stated objectives?

-Is the population clearly described and appropriate for the hypothesis being tested?

-Is the sample size sufficient to ensure adequate power to address the hypothesis being tested?

-Were correct statistical analysis used to support conclusions?

-Are there concerns about ethical or regulatory requirements being met?

Reviewer #1: I had no prior concerns about the methods and this remains the case

Reviewer #2: The authors explained the inclusion criteria in their response to reviewers but need to include it in the Methods section

**Results**

-Does the analysis presented match the analysis plan?

-Are the results clearly and completely presented?

-Are the figures (Tables, Images) of sufficient quality for clarity?

Reviewer #1: I appreciate that the authors have provided further information regarding steroid dosing in Table 2 with median and IQR, as this certainly gives a better sense than what is described in the body of the paper. I think adding this information in Table 1 for the non-fatal and fatal groups would also be useful though in order to have some sense as to whether those that died received a different sort of dosing regimen than those that survived. 

For Table 7, in their response regarding timing of complications relative to timing steroid dosing, they stated that due to the retrospective nature of the study, they had difficulty in determining timing of complications. This is understandable, but then this uncertainty should be made explicitly clear in the paper.

Reviewer #2: Overall much improved

The methods of testing (PCR and serology) were explained in their response to reviewers but were not included in the text.

The authors explained in their responses that the steroid dosages and types of steroids used differed significantly among centers but they need to add a comment in the text or as a footnote

**Conclusions**

-Are the conclusions supported by the data presented?

-Are the limitations of analysis clearly described?

-Do the authors discuss how these data can be helpful to advance our understanding of the topic under study?

-Is public health relevance addressed?

Reviewer #1: Following the point about timing of complications in results, the authors on pg 19, lines 328-329 state "In our study we observed an increase in various complications following steroid therapy"; there is a similar sentence on line 338. However, their response mentioned above seems to suggest that they were unable to actually assess the timing of most complications relative to steroid therapy. Thus, if they wish to point out the increase in complications, it should read something to the effect of "an increase in various complications in those who received steroid therapy". 

Otherwise, I believe their conclusions are adequately supported by the results. The most significant limitations of the analysis are beyond the control of the authors, namely sample size and heterogeneity of practice, but as they state, a prospective analysis would be needed to address this. Thus, demonstrating the uncertainty of the role of steroids and need for prospective study is perhaps the most important conclusion.

Reviewer #2: Limitations are explained better

**Editorial and Data Presentation Modifications?**

Reviewer #1: Minor revision and would endorse accepting after the above changes are made.

Reviewer #2: (No Response)

**Summary and General Comments**

Reviewer #1: The authors have incorporated most of the feedback previously provided and I believe the paper is stronger as a result. With a few small modifications and clarifications, I believe it would be suitable for publication. Although its small sample size and retrospective nature is certainly not enough to base practice on, it does provide some suggestion of worse outcome with steroids, in contradiction to other published results in support of steroids, and thus as the authors state, supports the need for prospective study of the subject.

Reviewer #2: I am satisfied with the authors responses overall

PLOS authors have the option to publish the peer review history of their article (what does this mean?). If published, this will include your full peer review and any attached files.

Reviewer #1: No

Reviewer #2: Yes: Ian C. Michelow, MD DTM&H
---

## [Editor Report · Decision Letter 2]

12 Jan 2021

Dear Dr. Kim,

We are pleased to inform you that your manuscript 'Effects of Steroid Therapy in Patients with Severe Fever with Thrombocytopenia Syndrome: A Multicenter Clinical Cohort Study' has been provisionally accepted for publication in PLOS Neglected Tropical Diseases.

Best regards,

Anita K. McElroy, MD, PhD

Associate Editor

A. Desiree LaBeaud

Deputy Editor

---

## [Editor Report · Acceptance letter]

16 Feb 2021

Dear Dr. Kim,

We are delighted to inform you that your manuscript, "Effects of Steroid Therapy in Patients with Severe Fever with Thrombocytopenia Syndrome: A Multicenter Clinical Cohort Study," has been formally accepted for publication in PLOS Neglected Tropical Diseases.

Best regards,

Shaden Kamhawi

co-Editor-in-Chief

Paul Brindley

co-Editor-in-Chief
